# Aorto-Iliac Artery Calcification and Graft Outcomes in Kidney Transplant Recipients

**DOI:** 10.3390/jcm10020325

**Published:** 2021-01-17

**Authors:** Stan Benjamens, Saleh Z. Alghamdi, Elsaline Rijkse, Charlotte A. te Velde-Keyzer, Stefan P. Berger, Cyril Moers, Martin H. de Borst, Riemer H. J. A. Slart, Frank J. M. F. Dor, Robert C. Minnee, Robert A. Pol

**Affiliations:** 1Department of Surgery, Division of Transplant Surgery, University of Groningen, University Medical Center Groningen, 9713 GZ Groningen, The Netherlands; s.z.alghamdi@student.rug.nl (S.Z.A.); c.moers@umcg.nl (C.M.); r.pol@umcg.nl (R.A.P.); 2Medical Imaging Center, Department of Nuclear Medicine and Molecular Imaging, University of Groningen, University Medical Center Groningen, 9713 GZ Groningen, The Netherlands; r.h.j.a.slart@umcg.nl; 3Department of Surgery, Division of HPB and Transplant Surgery, Erasmus MC University Medical Center, 3015 CE Rotterdam, The Netherlands; a.rijkse@erasmusmc.nl (E.R.); r.minnee@erasmusmc.nl (R.C.M.); 4Department of Internal Medicine, Division of Nephrology, University of Groningen, University Medical Center Groningen, 9713 GZ Groningen, The Netherlands; c.a.keyzer@umcg.nl (C.A.t.V.-K.); s.p.berger@umcg.nl (S.P.B.); m.h.de.borst@umcg.nl (M.H.d.B.); 5Department of Biomedical Photonic Imaging, Faculty of Science and Technology, University of Twente, 7522 NB Enschede, The Netherlands; 6Imperial College Renal and Transplant Centre, Hammersmith Hospital, Imperial College Healthcare NHS Trust, London W12 0HS, UK; frank.dor@nhs.net; 7Department of Surgery & Cancer, Imperial College, London SW7 2BU, UK

**Keywords:** kidney transplantation, vascular calcification, aorta, iliac artery, graft function, graft failure, graft function decline

## Abstract

While the association of vascular calcification with inferior patient outcomes in kidney transplant recipients is well-established, the association with graft outcomes has received less attention. With this dual-centre cohort study, we aimed to determine the clinical impact of recipient pre-transplant aorto-iliac calcification, measured on non-contrast enhanced computed tomography (CT)-imaging within three years prior to transplantation (2005–2018). We included 547 patients (61.4% male, age 60 (interquartile range 51–68) years), with a median follow-up of 3.1 (1.4–5.2) years after transplantation. The aorto-iliac calcification score (CaScore) was inversely associated with one-year estimated-glomerular filtration rate (eGFR) in univariate linear regression analysis (standard β −3.3 (95% CI −5.1 to −1.5, *p* < 0.0001), but not after adjustment for potential confounders, including donor and recipient age (*p* = 0.077). In multivariable Cox regression analyses, a high CaScore was associated with overall graft failure (*p* = 0.004) and death with a functioning graft (*p* = 0.002), but not with death-censored graft failure and graft function decline. This study demonstrated that pre-transplant aorto-iliac calcification is associated with one-year eGFR in univariate, but not in multivariable linear regression analyses. Moreover, this study underlines that transplantation in patients with a high CaScore does not result in earlier transplant function decline or worse death censored graft survival, although ongoing efforts for the prevention of death with a functioning graft remain essential.

## 1. Introduction

Atherosclerosis is an important pathophysiological process responsible for an increased disease burden in kidney transplant candidates and recipients, and is associated with decreased overall survival and increased cardiovascular morbidity and mortality [1,2,3]. Currently, there is limited published data on the effect of recipient pre-transplant aorto-iliac calcification on kidney graft outcomes.

Kidney graft outcomes can be divided in short-term outcomes, such as early failure or one-year graft function, and long-term outcomes, such as (death-censored) graft failure and graft function decline [4]. Death-censored graft failure, as return to dialysis or re-transplantation, is seen in 10–12.5% of kidney transplant recipients at 5–6.2 years after transplantation [5,6]. Overall graft failure, including death with a functioning graft, is seen at a rate of 5% each year of follow-up, of which 40–60% is attributed to death with a functioning graft [7,8,9,10,11]. Graft function decline, as doubling of serum creatinine or graft failure, is seen in 18.8% of kidney transplant recipients at 5.4 years after transplantation [12]. Short and long-term graft outcomes are strongly associated and both closely linked to patient-specific and transplantation-related risk factors [13,14,15,16].

A recent meta-analysis on pre-transplant aorto-iliac calcification found an association with decreased overall survival and overall graft failure. Statistical significance was lacking for the association with death-censored graft failure and graft function. This lack of statistical power with regard to kidney graft outcomes can be attributed to the limited number of studies available, the various methods used for identification of calcification and the small sample sizes of the existing publications [17]. Hence, studies including higher numbers of transplant recipients, focusing on various kidney graft outcomes and applying reliable methods of calcifications assessment, are deemed pivotal.

We previously reported that the aorto-iliac calcification score (CaScore), the adjusted Agatston score, was independently associated with patient outcomes, e.g., risk of early (cardiovascular) death and cardiovascular events [1]. In the present study, we aimed to determine whether the aorto-iliac CaScore, measured as an adjusted Agatston score on pre-transplant computed tomography (CT)-imaging, is associated with kidney graft function at one-year after transplantation, death-censored graft failure, overall graft failure (including death), death with a function graft, and graft function decline.

## 2. Methods

### 2.1. Study Design

We performed a dual-centre cohort study in The Netherlands (University Medical Center Groningen (UMCG) and Erasmus University Medical Center (Erasmus MC)), in adult kidney transplant recipients. All patients who underwent screening prior to transplantation by means of a non-contrast enhanced CT between January 2005 and December 2018 were included. CT procedures were performed according to the pre-transplant screening protocol in both transplant centres; selecting patients based on age (>50 years); comorbidities (diabetes mellitus and peripheral artery disease) and dialysis vintage (>2 years). Exclusion criteria were age <18 years at the time of transplantation and an interval of more than three years between CT and transplantation. In the inclusion period, both transplant centres participated in the Eurotranplant Senior Program (ESP), a Eurotransplant allocation scheme matching deceased donors above 65 years-of-age to recipients in a similar age range [18]. The primary outcome measure of this study was estimated-glomerular filtration rate (eGFR) at one-year after transplantation. Secondary outcome measures were death-censored graft failure, defined as return to dialysis or re-transplantation (censoring at time of death), overall graft failure (including death with a functioning graft), death with a functioning graft, and graft function decline, defined as doubling in serum creatinine or graft failure (censoring at time of death). This study was approved by the institutional review board of the UMCG (2017/523) and performed in line with the Declaration of Helsinki and the Declaration of Istanbul on Organ Trafficking and Transplant Tourism.

### 2.2. Computed Tomography (CT) Assessment of Aorto-Iliac Calcification

The applied method for quantifying aorto-iliac calcification, as the Agatston score adjusted for the aorto-iliac trajectory (aorto-iliac CaScore) for non-enhanced CT images, has been described previously [1]. In short, CaScore (syngo.CT CaScoring software, Siemens Healthineers, Erlangen, Germany) was quantified for the abdominal aorta below the origin of the renal arteries, the common iliac artery, and the external iliac artery for the side of the transplant anastomosis. The standard calcification threshold of 130 Hounsfield units (HU) was used.

### 2.3. Clinical Variables

Details of this cohort, including baseline cardiovascular risk factors, with history and follow-up of cardiovascular disease, have been published previously [1]. The recipient variables included in the present analyses were gender, age, pre-transplant diabetes mellitus (fasting glucose ≥7.0 mmol/L, casual plasma glucose ≥11.1 mmol/L + diabetes symptoms, or glucose tolerance test with 2-h plasma glucose ≥11.1 mmol/L), body mass index (BMI in kg/m^2^), smoking status (non, former, or current), hypercholesterolemia (total cholesterol level >5.2 mmol/L or use of lipid lower medication), total cholesterol level (mmol/L), systolic blood pressure (mmHg), use of antihypertensive medication, dialysis status (pre-emptive, haemodialysis, or peritoneal dialysis), dialysis vintage (months), and number of previous kidney transplantations. The transplant variables included the type of donation (living-donation, donation after circulatory death (DCD), or donation after brain death (DBD)), donor gender and age, number of total human leukocyte antigen (HLA)-mismatches, warm ischemia time (minutes), and cold ischemia time (minute). Follow-up variables included eGFR at six-months and one-year, calculated with the Chronic Kidney Disease Epidemiology Collaboration (CKD-EPI) equation (mL/min per 1.73 m^2^), laboratory values at one-year after transplantation (serum haemoglobin, calcium, phosphate, albumin, glucose, parathyroid hormone (PTH), and urinary protein excretion), and primary cytomegalovirus (CMV) infection [19]. Short-term outcomes were early graft failure (death-censored), defined as return to dialysis in 30-days after transplantation, delayed graft function, defined as dialysis requirement in the first week after transplantation, acute rejection, defined as biopsy-proven acute rejection (BPAR) or an acute rejection treatment episode without BPAR (non-BPAR, Banff 2015 criteria) [4,20].

### 2.4. Statistical Analyses

For baseline characteristics, patients were stratified into two groups based on the aorto-iliac CaScore, being a low CaScore (0–5600 HU) and a high CaScore (>5600 HU). These cut-off values are based on the initial aorto-iliac CaScore analyses, with the high CaScore derived from the highest tertile of CaScores [1]. Data were expressed as mean (standard deviation, SD) for normally distributed variables, as median (interquartile range, IQR) for variables with non-normal distribution, and as number (percentage, %) for categorical variables. For regression analyses, non-normally distributed variables were transformed to the natural log (ln (x + 1)). Results of linear regression were presented as standardized Beta coefficients (Standard β) and in proportional hazards regression analysis as hazard ratios (HRs), both with 95% confidence intervals (95%CI). The reversed Kaplan-Meier method was used to calculate the median (interquartile range (IQR)) follow-up, considering the date of transplantation as the start of follow-up [21]. In all analyses, a *p*-value <0.05 was considered significant.

(Death-censored) graft failure-free and graft function decline-free were calculated using Kaplan-Meier survival curves. The low and high aorto-iliac CaScores were compared for living and deceased-donor kidney transplant recipients with Log-rank testing. Cox proportional hazards regression analysis, for living and deceased-donor kidney transplant recipients, was used to establish the association of the high aorto-iliac CaScore with (death-censored) graft failure, death with a functioning graft and graft function decline. Linear regression was used to establish the association of the aorto-iliac CaScore (continuous) with one-year eGFR. Models were built with a priori selected covariables. In Model 1, we adjusted for transplant centre (UMCG or Erasmus MC) and time between CT and transplantation. In model 2, we adjusted for model 1 plus donor gender, donor type (living donation, DCD or DBD), cold ischemia time, number of HLA mismatches, recipient gender, diabetes mellitus, smoking, dialysis vintage, number of previous transplantations, statin use pre-transplantation. In model 3, we adjusted for model 2 plus recipient age and in model 4 for model 3 plus donor age. For linear regression, laboratory results at one year after transplantation (serum phosphate, serum calcium, serum glucose, serum haemoglobin, serum PTH, proteinuria) and an episode of acute rejection were included as additional variables in model 2. Analysis of interaction terms for the association of the aorto-iliac CaScore (continuous) with one-year eGFR by the covariables recipient age, gender, diabetes, donor age, gender, and donor type, was performed. To quantify the magnitude of confounding by donor age on the association between the aorto-iliac CaScore and one-year eGFR, we used the regression methodology as proposed by Janes et al. [22]. In addition, linear regression was performed after multiple imputation by chained equations (MICE), using predictive mean matching and five imputed datasets, with imputations for serum PTH and proteinuria.

Statistical analyses were performed with R: A Language and Environment for Statistical Computing, version 1.0.153 for Mac (R Foundation for Statistical Computing, Vienna, Austria), using the software R-Packages “MASS”, “MICE” “mediation”, “survival”, “ggplot2”, and “survminer”.

## 3. Results

### 3.1. Characteristics and Short-Term Graft Outcomes

A total of 547 patients were included, of which 446 in the UMCG and 101 in the Erasmus MC (61.4% male, age 60 (51, 68) years). The mean time from CT to transplantation was 0.91 (0.72) years and the median follow-up was 3.1 (1.4, 5.2) years after transplantation. Appendix A shows the numbers for death, graft failure and end of follow-up in the first three years after kidney transplantation. Baseline patient characteristics for the total cohort and CaScore groups are shown in Table 1. In the high compared to the low CaScore group, transplant recipients were older (66 (60, 71) vs. 55 (47, 64)) (Appendix A), patients had a higher donor age (59 (12) vs. 52 (14) years), there were fewer living-donor (47.3% vs. 58.9%), and more DCD (29.1% vs. 20.5%) and DBD (23.6% vs. 20.5%) transplantation procedures. In the first year of follow-up, laboratory values (*n* = 462) of the CaScore groups differed for eGFR at six-months and one-year, calcium, phosphate, glucose, PTH, and urinary protein excretion (Table 1). For short-term graft outcomes, more early graft failure (death-censored) events were observed in the high compared to the low CaScore group (*n* = 9 (4.9) vs. *n* = 5 (1.4), *p* = 0.027), but no differences in delayed graft function or acute rejections (Table 2). In nine out of the 14 patients with early graft failure, graft failure was classified as due to a vascular cause. In Appendix A, patient characteristics are presented for living and deceased donation separately.

### 3.2. Aorto-Iliac Calcification Score (CaScore) and One-Year Kidney Transplant Function

Mean eGFR at one-year after transplantation (*n* = 462) was 51 (21), with 48 (21) in the high and 53 (20) in the low CaScore group. The CaScore (continuous) was inversely associated with one-year eGFR, with a standard β −3.5 (95% CI −5.3 to −1.7, *p* < 0.0001) when adjusting for transplant centre and time CT to transplantation (model 1). When adjusting for patient, transplantation and follow-up covariables, with the exception of donor age, this inverse association remained significant (model 2, standard β −4.1 (95% CI −4.9 to −2.4, *p* < 0.0001), and model 3 (including recipient age), standard β −2.7 (95% CI −4.6 to −0.7, *p* = 0.008)). This association lost statistical significance after adjustment for donor age (model 4, standard β −1.7 (95% CI −3.6 to 0.2, *p* = 0.077)) (Table 3). There was a significant interaction for the univariable association of CaScore with one-year eGFR by donor age (*p* < 0.0001), but not for the covariables recipient age, gender, diabetes, donor gender, and donor type (*p* > 0.05). When quantifying the magnitude of confounding, the direct effect of the aorto-iliac CaScore on one-year eGFR was −3.3 (95% CI −5.1 to −1.5) and the effect on the confounder (donor age) was 5.0 (95% CI 3.8 to 6.2). The direct effect of the confounder on one-year eGFR was −0.6 (95% CI −0.7 to −0.5), with a bootstrapped indirect effect of −2.9 (95% CI −4.0 to −1.9). Donor age was identified as a confounder of the association between the aorto-iliac CaScore and one-year eGFR, resulting in an 87.2% of the association (Figure 1). In Appendix A, the linear regression for one-year eGFR is presented for the MICE model. In Appendix A, the linear regression in presented for living- and deceased donation separately.

### 3.3. Aorto-Iliac CaScore and Death-Censored Graft Failure

A total of 64 (11.7%) events of death-censored graft failure occurred, with 23 (12.6%) in the high and 41 (11.2%) in the low CaScore group (Table 2). Kaplan–Meier survival analysis for death-censored graft failure, with the low and high CaScore groups stratified by living and deceased donor kidney transplantation, showed earlier graft failure for the deceased donor—high CaScore group (*p* < 0.001) (Figure 2). Kaplan–Meier survival analysis stratified by transplant recipient age, below or above 65 years-of-age, did not show significant differences (*p* = 0.078, Appendix A). In Cox regression analysis, no significant associations were observed for a high CaScore and death-censored graft failure (model 4, HR 1.1 (95% CI 0.6 to 2.0, *p* = 0.711)) (Table 4). No differences were observed when stratifying for living- and deceased donation (model 4, Appendix A).

### 3.4. Aorto-Iliac CaScore and Overall Graft Failure

A total of 118 (21.6%) events of overall graft failure occurred, with 54 (29.7%) in the high and 64 (17.5%) in the low CaScore group (Table 2). Kaplan–Meier survival analysis for overall graft failure, with the low and high CaScore group stratified by living and deceased donor kidney transplantation, showed earlier graft failure for the living and deceased donor—high CaScore groups (*p* < 0.0001) (Figure 2). Kaplan–Meier survival analysis stratified by transplant recipient age showed, a lower survival for the recipients ≥65 years-of-age—high CaScore group (*p* < 0.0001) (Appendix A). In Cox regression analysis, a significant association was observed for a high CaScore and graft failure after adjusting for patient and transplantation covariables, including recipient and donor age (model 4, HR 1.9 (95% CI 1.2 to 2.9, *p* = 0.006)) (Table 4). When stratifying for donor status, the association remained significant for deceased donation (model 4, *p* = 0.048) and for living-donation (model 4, *p* = 0.0036) (Appendix A).

### 3.5. Aorto-Iliac CaScore and Death with a Functioning Graft

A total of 54 (9.9%) events of death with a functioning graft occurred, with 31 (17.0%) in the high and 23 (6.3%) in the low CaScore group (Table 2). Kaplan–Meier survival analysis for death with a functioning graft, with the low and high CaScore group stratified by living and deceased donor kidney transplantation, showed earlier graft failure for the living and deceased donor—high CaScore groups (*p* < 0.0001) (Figure 3). Kaplan-Meier survival analysis stratified by transplant recipient age showed, a lower survival for the recipients ≥65 years-of-age—high CaScore group (*p* < 0.0001) (Appendix A). In Cox regression analysis, a significant association was observed for a high CaScore and death with a functioning graft after adjusting for patient and transplantation covariables, including recipient and donor age (model 4, HR 2.7 (95% CI 1.4 to 5.0, *p* = 0.224)) (Table 4). When stratifying for donor status, the association remained significant for both living- and deceased-donation (model 4, *p* = 0.015 and *p* = 0.042, respectively) (Appendix A).

### 3.6. Aorto-Iliac CaScore and Graft Function Decline

A total of 81 (14.8%) events of graft function decline occurred, with 51 (14.0%) in the high and 30 (16.5%) in the low CaScore group (Table 2). Kaplan–Meier survival analysis for graft function decline, with the low and high CaScore group stratified by living and deceased donor kidney transplantation, showed a lower time-to-graft function decline for deceased donor—high CaScore group (*p* < 0.0001) (Figure 4). Kaplan–Meier survival analysis stratified by transplant recipient age, showed a lower survival for the recipients ≥65 years-of-age—high CaScore group (*p* = 0.014) (Appendix A). In Cox regression analysis, no significant associations were observed between the high CaScore and graft function decline after adjusting for patient and transplantation covariables, including recipient and donor age (model 4, HR 1.4 (95% CI 0.8 to 2.4, *p* = 0.223) (Table 4). No differences were observed when stratifying for living- and deceased donation (model 4, Appendix A).

## 4. Discussion

In this dual centre cohort study, we identified a non-causal inverse association between pre-transplant aorto-iliac calcification and one-year eGFR in univariate analysis. This association was not significant after adjustment for patient-specific, transplantation-related, and follow-up covariables, including donor and recipient age. The increased donor age in older recipients, due to the allocation of older donor kidneys to older recipients in the Eurotransplant Senior Program, was a significant confounder. For long-term graft outcomes, no association could be established between a high aorto-iliac CaScore and an increased risk of death-censored graft failure or graft function decline, in both living and deceased donor kidney transplantation. Whereas the identified association between a high aorto-iliac CaScore and graft failure (death-censored) remained significant after adjustment for various covariables, including donor and recipient age. Also, the high aorto-iliac CaScore was associated with death with a functioning graft.

Our study has both similarities and important differences compared to four previously published studies on pre-transplant imaging-based visualization of calcification. In the largest study (*n* = 374), aorto-iliac stenosis on CT-imaging was not an independent risk factor for graft failure, but was associated with an increased risk of overall graft failure [23]. In a relatively small study (*n* = 119) with lateral lumbar radiography-based calcification assessment, transplant recipients with and without calcifications differed with regard to graft failure-free survival [24]. In a study on pre-transplant CT-based quantification of calcification (*n* = 100), no statistically significant difference in eGFR at one-year after transplantation was found for the quartiles of aorto-iliac calcification. Also, survival analysis for graft failure, with nine events after a median follow-up of 2.6 years, did not show differences between the quartiles [25]. In a cohort of 93 transplant recipients, Aitken et al. showed that visual two group stratification of vascular calcification on pelvic X-ray did result in graft survival differences, but not in differences for one-year creatinine [26]. Compared to these imaging studies, the current study has the largest sample size and a relatively long period of follow-up. The results of our study were comparable for living- and deceased-donation, with an exception for the outcome measure graft function decline. In univariate analyses, there was a significant association between the CaScore and graft function decline for deceased-donation, whereas the association was not significant for living-donation. A possible explanation is the significant age difference and difference in diabetes mellitus prevalence between the living- and deceased-donation patients.

The outcomes of our study are in line with previous non-imaging studies on calcification. In a histology-based cohort study (*n* = 90), differences in graft failure were observed for transplant recipients with and without intimal microcalcification of the iliac artery; however, these differences were not tested in multivariable analyses. For graft function decline, an independent association with the combination of intimal microcalcification with positive intimal fetuin-A was identified [27]. In post-transplant blood serum analysis (*n* = 699), calciprotein particle maturation time (T50), a measure for blood calcification propensity, were shown to be associated with all-cause mortality, cardiovascular mortality, and graft failure [28].

A global development in transplant programs is a steady increase in donor and transplant candidate age [29]. With this trend, the causal relation of recipient age with vascular calcification and the confounding effect of donor age on the association between calcification and outcomes, can be important factors affecting transplant outcomes. Efforts could go to the optimization of kidney function and patient survival for transplant recipients with a high degree of aorto-iliac calcification. The kidney function of grafts from older donors may be improved, by decreasing cold ischemia time, reducing kidney rewarming during transplantation, and optimizing the use of hypothermic and normothermic machine perfusion [30,31,32,33]. To improve patient survival in general and death with a functioning graft specially, focus might go to the reduction of modifiable risk factors for cardiovascular disease in kidney transplant candidates and recipients. This risk reduction, pre- and post-transplantation, could be achieved by focusing on prevention of new-onset diabetes, preventive measures for mineral and bone disorders, and reduction of hyperlipidaemia and hypertension [34]. Medical strategies in line with these focus points are age-adapted immunosuppressive therapy, stringent use of statins and antihypertensive medication, and general lifestyle interventions [35,36,37].

Our study has a few limitations that need to be addressed. First, we retrospectively selected patients in which a CT procedure was already performed based on predefined criteria, making the study prone to selection bias and confounding. Although the presented findings are applicable to the majority of transplant recipients with comparable baseline characteristics (median age of 60 (51–68) years), they should not be extrapolated to relatively young patients with limited comorbidities. Second, the two transplant centres in this cohort study are part of Eurotransplant Senior Program [18,38]. This particular allocation scheme could have influenced the association between the aorto-iliac CaScore and donor age, given the causal relationship between vascular calcification and recipient age. Third, the period of follow-up of 3.1 years (median) is relatively long compared to previous imaging studies on vascular calcification but on the short side compared to large registry studies on the topic of kidney graft outcomes. Fourth, for a relatively large proportion of patients (*n* = 85; of which 31 patients experienced overall graft failure with the first year) laboratory values at one-year after transplantation were not available. Complying with general standards for dealing with missing data, a complete case analysis was found to be most reliable. In addition, linear regression was performed after multiple imputation, with imputations for serum PTH and proteinuria. Fifth, the interpretation of the implications of a statistically significant difference should always include a reflection of the clinical significance of the observed difference. In this study, a mean difference in eGFR of 5 mL/min per 1.73 m^2^ was found between a high and a low aorto-iliac CaScore (mean 48 (21) vs. 53 (20)). With a mean eGFR of 51 in the total cohort, this is a 10% difference in one-year graft function.

Nonetheless, this study has its unique strengths, which justify attention. This is the first study evaluating the association between aorto-iliac calcification and one-year graft function in multivariable analysis, including patient, transplantation, and follow-up covariables. Second, the aorto-iliac CaScore is an easy-to-implement assessment, as the software is readily available in standard imaging analysis packages and pre-transplant CT-imaging is advised for high-risk transplant candidates [39,40]. Third, the discrepancy between the results of Kaplan–Meier survival analysis and multivariable Cox regression analyses highlights the importance of relatively large cohort studies that enable adjustment for important covariables. Kaplan–Meier survival analysis indicated significant differences based on the aorto-iliac CaScore for (death-censored) graft failure and graft function decline, whereas a significant independent association in Cox regression analysis was only found for overall graft failure and death with a function graft.

## 5. Conclusions

In conclusion, this study demonstrated that pre-transplant aorto-iliac calcification is associated with one-year eGFR in univariate, but not in multivariable linear regression analyses. The increased donor age in older recipients is a significant confounder for this association. A high CaScore was not associated with death-censored graft failure and graft function decline, whereas independent associations with overall graft failure and death with a functioning graft were identified. These results underline that transplantation in patients with a high CaScore does not result in earlier transplant function decline or worse death censored graft survival, although ongoing efforts for the prevention of death with a functioning graft remain essential.

## Figures and Tables

**Figure 1 jcm-10-00325-f001:**
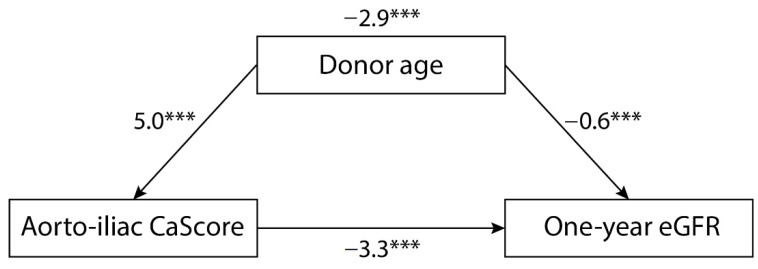
Graphical illustration of the magnitude of confounding for one-year estimated glomerular filtration rate (eGFR), showing the regression coefficients for the associations (*** indicating a *p*-value < 0.0001). The direct effect of the aorto-iliac CaScore on one-year eGFR was −3.3 (95%CI −5.1 to −1.5) and the effect on the confounder (donor age) was 5.0 (95%CI 3.8 to 6.2). The direct effect of the confounder on one-year eGFR was −0.6 (95%CI −0.7 to −0.5), with a bootstrapped indirect effect of −2.9 (95% CI −4.0 to −1.9).

**Figure 2 jcm-10-00325-f002:**
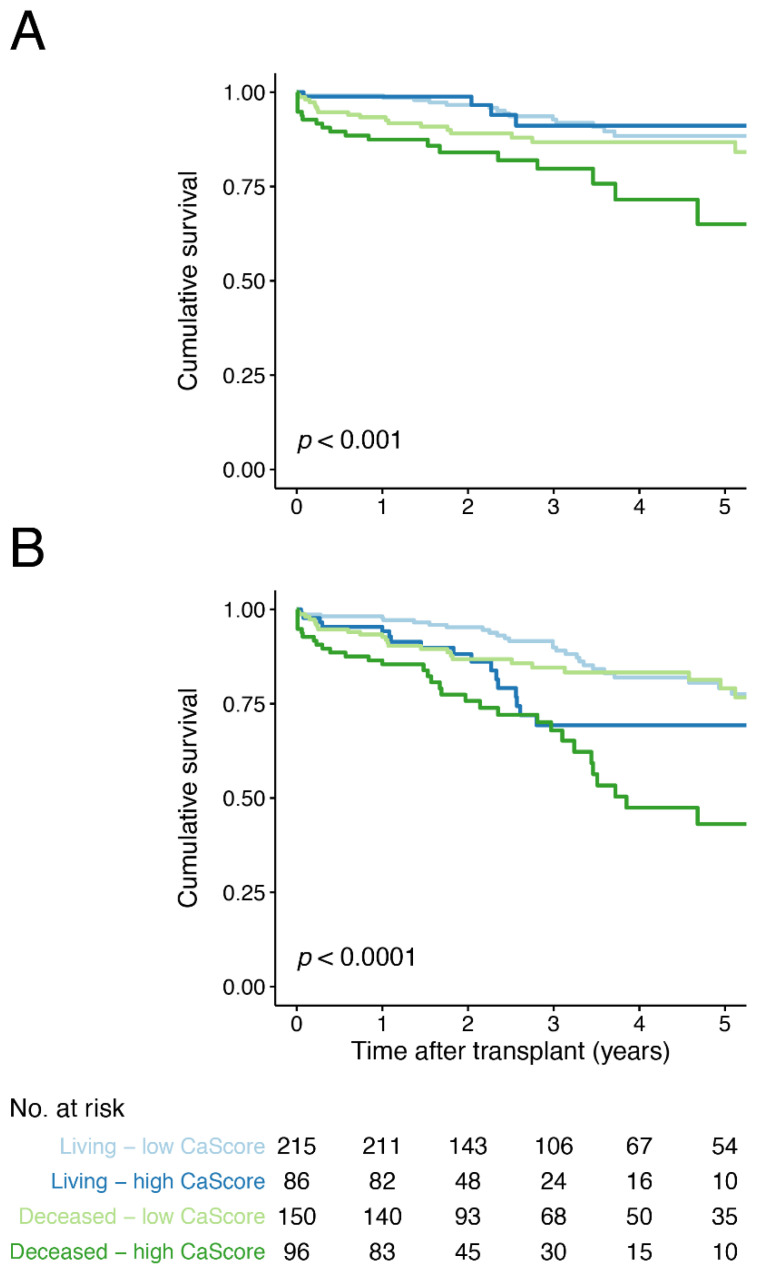
Kaplan–Meier survival curve for (**A**) death-censored and (**B**) overall graft failure free-survival for the low and high aorto-iliac CaScore group, stratified by living and deceased donor kidney transplantation and including a life table applicable for both graphs. Number at risk (No. at risk) provided for all four groups.

**Figure 3 jcm-10-00325-f003:**
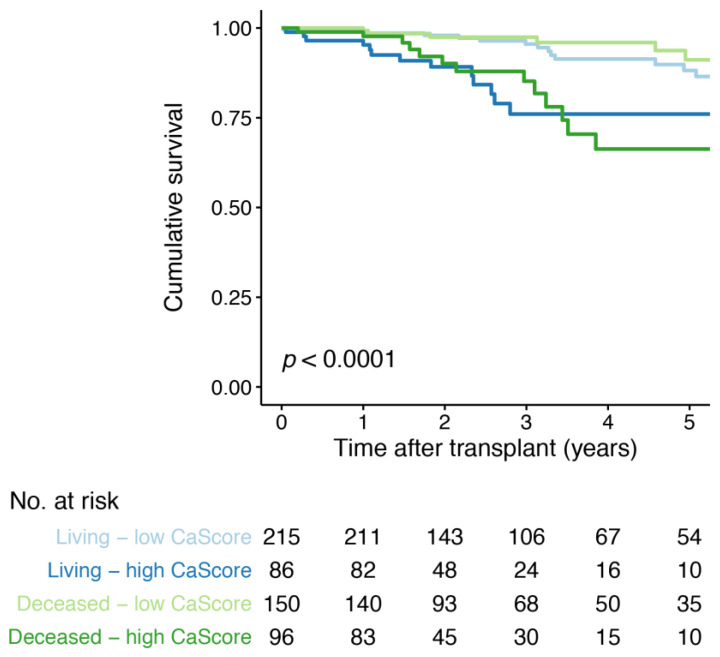
Kaplan–Meier survival curve for death with a functioning graft free-survival for the low and high aorto-iliac CaScore group, stratified by living and deceased donor kidney transplantation. Number at risk (No. at risk) provided for all four groups.

**Figure 4 jcm-10-00325-f004:**
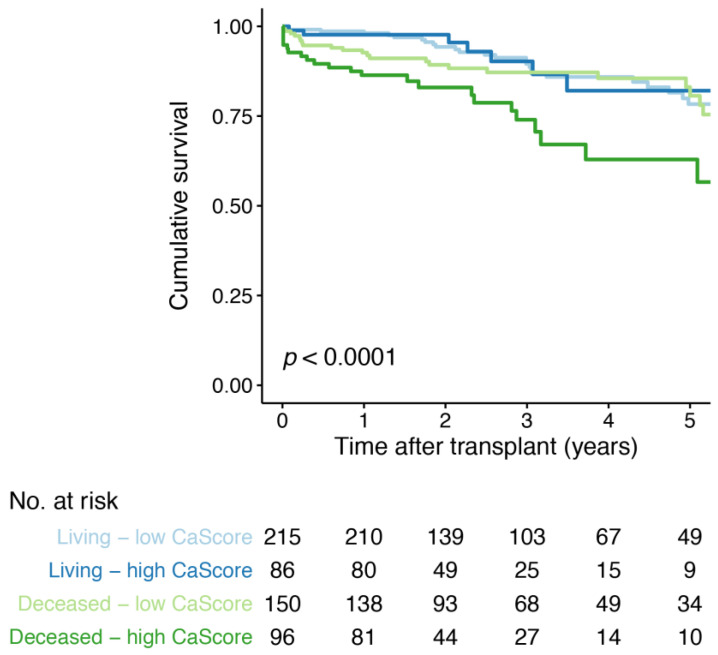
Kaplan–Meier survival curve for graft function decline free-survival for the low and high aorto-iliac CaScore group, stratified by living and deceased donor kidney transplantation. Number at risk (No. at risk) provided for all four groups.

**Table 1 jcm-10-00325-t001:** Characteristics.

Variables	Total (*n* = 547)	Low CaScore (*n* = 365)	High CaScore (*n* = 182)	*p*-Value
**Patient**				
Male gender, *n* (%) ^a^	336 (61.4)	205 (56.2)	131 (72.0)	<0.001 ^b^
Age, years ^c^	60 (51, 68)	55 (47, 64)	66 (60, 71)	<0.001 ^d^
Diabetes mellitus, *n* (%) ^a^	173 (31.6)	117 (32.1)	56 (30.8)	0.836 ^b^
Body Mass Index, kg/m^2 b^	26.8 (4.7)	26.9 (4.9)	26.7 (4.1)	0.670 ^c^
Smoker, *n* (%) ^a^				0.004 ^b^
Non	154 (28.2)	119 (32.6)	35 (19.2)	
Former	332 (60.7)	206 (56.4)	126 (69.2)	
Current	61 (11.2)	40 (11.0)	21 (11.5)	
Hypercholesterolemia, *n* (%) ^a^	153 (28.0)	96 (26.3)	57 (31.3)	0.258 ^b^
Total cholesterol, mmol/L	4.7 (1.3)	4.8 (1.5)	4.5 (1.2)	0.031 ^c^
Systolic blood pressure, mmHg	145 (22)	143 (21)	148 (24)	0.014 ^c^
Use of antihypertensive medication, *n* (%)	439 (80.3)	282 (77.3)	157 (86.3)	0.017 ^b^
Type of dialysis, *n* (%) ^a^				0.109 ^b^
Pre-emptive	197 (36.0)	142 (38.9)	55 (30.2)	
Haemodialysis	250 (45.7)	162 (44.4)	88 (48.4)	
Peritoneal dialysis	100 (18.3)	61 (16.7)	39 (21.4)	
Dialysis vintage, months ^c^	12 (0, 28)	11 (0, 25)	16 (0, 33)	0.012 ^d^
Previous transplants, *n* (%) ^a^				0.872 ^b^
Non	514 (94.0)	343 (94.0)	171 (94.0)	
One	22 (4.0)	14 (3.8)	8 (4.4)	
Two	11 (2.0)	8 (2.2)	3 (1.6)	
Aorto-iliac CaScore, HU	2994 (294, 7888)	856 (46, 2994)	9732 (7918, 14315)	<0.001 ^d^
**Transplantation**				
Type of donation, *n* (%) ^a^				0.025 ^b^
Living-donation	301 (55.0)	215 (58.9)	86 (47.3)	
Donation after circulatory death	128 (23.4)	75 (20.5)	53 (29.1)	
Donation after brain death	118 (21.6)	75 (20.5)	43 (23.6)	
Male gender donor, n (%)	253 (46.3)	160 (43.8)	93 (51.1)	0.130 ^b^
Donor age, years	54 (14)	52 (14)	59 (12)	<0.001 ^c^
No. of HLA-mismatches, *n*	3.5 (1.5)	3.4 (1.5)	3.6 (1.4)	0.127 ^c^
Warm ischemia time, minutes	42 (36)	42 (44)	41 (13)	0.676 ^c^
Cold ischemia time, minutes	433 (340)	415(343)	470 (331)	0.076 ^c^
**Follow-up ^e^**				
eGFR at six-months ^a^	50 (18)	51 (19)	46 (17)	0.004 ^c^
eGFR at one-year ^a^	51 (21)	53 (20)	48 (21)	0.007 ^c^
Haemoglobin at one-year, mmol/L	8.3 (3.8)	8.3 (4.6)	8.2 (1.1)	0.811 ^c^
Calcium at one-year, mmol/L	2.43 (0.14)	2.43 (0.14)	2.45 (0.14)	0.034 ^c^
Phosphate at one year, mmol/L	0.93 (0.21)	0.95 (0.21)	0.89 (0.19)	0.004 ^c^
Albumin at one-year, g/L	43 (3)	43 (3)	43 (3)	0.234 ^c^
Glucose at one-year, mmol/L	5.9 (5.1, 7.3)	5.7 (5.1, 7.2)	6.2 (5.4, 7.6)	0.005 ^d^
PTH at one-year, pmol/L	10 (7, 16)	10 (7, 15)	12 (8, 16)	0.064 ^d^
Protein excretion at one-year, g/24 h	0.2 (0.1, 0.3)	0.1 (0.1, 0.2)	0.2 (0.1, 0.3)	0.002 ^d^
Cytomegalovirus infection, n (%)	29 (5.3)	17 (4.7)	12 (6.6)	0.453 ^b^

HU = Hounsfield units; HLA = human leukocyte antigen; eGFR = estimated glomerular filtration rate; PTH = parathyroid hormone; ^a^ eGFR—CKD-EPI (Chronic Kidney Disease Epidemiology Collaboration) (mL/min per 1.73 m20; ^b^
*p*-value by chi-square test; ^c^
*p*-value by Student’s *t*-test; ^d^
*p*-value by Mann–Whitney U test; ^e^ data available for 462 patients.

**Table 2 jcm-10-00325-t002:** Short and long-term graft outcomes (numbers, %).

Outcome Measures	Total (*n* = 547)	Low CaScore (*n* = 365)	High CaScore (*n* = 182)	*p*-Value
**Median follow-up**, years	3.1 (1.4, 5.2)	3.2 (1.5, 5.6)	2.7 (1.2, 4.2)	
**Short-term graft outcomes**				
Early graft failure (death-censored)	14 (2.6)	5 (1.4)	9 (4.9)	0.027
Delayed graft function	155 (28.3)	100 (27.4)	55 (30.2)	0.555
Acute rejection (first-year)	70 (12.8)	47 (12.9)	23 (12.6)	1.000
**Long-term graft outcomes**				
Death-censored graft failure	64 (11.7)	41 (11.2)	23 (12.6)	0.734
Overall graft failure	118 (21.6)	64 (17.5)	54 (29.7)	0.002
Death with a functioning graft	54 (9.9)	23 (6.3)	31 (17.0)	<0.001
Graft function decline	81 (14.8)	51 (14.0)	30 (16.5)	0.515

Median follow-up after transplantation (interquartile range); number of short-term and long-term graft outcomes (%); *p*-value by chi-square test.

**Table 3 jcm-10-00325-t003:** Multivariable adjusted associations of the CaScore with one-year eGFR.

One-Year eGFR	Standard β	95% CI	*p*-Value
Univariate	−3.3	−5.1 to −1.5	<0.0001
Model 1	−3.5	−5.3 to −1.7	<0.0001
Model 2	−4.1	−5.9 to −2.4	<0.0001
Model 3	−2.7	−4.6 to −0.7	0.008
Model 4	−1.7	−3.6 to 0.2	0.077

Linear regression analysis, data available for 462 patients. Data are presented as hazard ratio and 95% confidence interval (CI) for the continuous CaScore (natural log transformed). Model 1: adjusted for transplant centre and time between computed tomography and transplantation; model 2: adjusted for model 1 plus donor gender, donor type (living donation, donation after circulatory death (DCD) or donation after brain death (DBD)), cold ischemia time, no. of human leukocyte antigen (HLA) mismatches, recipient gender, diabetes mellitus, smoking, dialysis vintage, no. of previous transplantations, statin use, laboratory results at one year after transplantation (serum phosphate, serum calcium, serum glucose, serum haemoglobin, serum PTH, proteinuria), and episode of acute rejection; model 3: adjusted for model 2 plus recipient age; model 4: adjusted for model 3 plus donor age.

**Table 4 jcm-10-00325-t004:** Multivariable adjusted associations of the CaScore with (death-censored) graft failure, death with a functioning graft and graft function decline.

Outcomes Measures	Low CaScore	High CaScore		
	Hazard Ratio	Hazard Ratio	95% CI	*p*-Value
**Death censored graft failure**				
Univariate	1.0 (Ref)	1.5	0.9 to 2.5	0.125
Model 1	1.0 (Ref)	1.6	0.9 to 2.7	0.082
Model 2	1.0 (Ref)	1.4	0.8 to 2.4	0.211
Model 3	1.0 (Ref)	1.1	0.6 to 2.0	0.697
Model 4	1.0 (Ref)	1.1	0.6 to 2.0	0.711
**Overall graft failure**				
Univariate	1.0 (Ref)	2.4	1.7 to 3.5	<0.0001
Model 1	1.0 (Ref)	2.5	1.7 to 3.7	<0.0001
Model 2	1.0 (Ref)	2.4	1.6 to 3.5	<0.0001
Model 3	1.0 (Ref)	1.8	1.2 to 2.7	0.008
Model 4	1.0 (Ref)	1.9	1.2 to 2.9	0.006
**Death with a functioning graft**				
Univariate	1.0 (Ref)	3.8	2.2 to 6.5	<0.0001
Model 1	1.0 (Ref)	3.9	2.3 to 6.9	<0.0001
Model 2	1.0 (Ref)	4.0	2.3 to 7.1	<0.0001
Model 3	1.0 (Ref)	2.8	1.5 to 5.2	0.002
Model 4	1.0 (Ref)	2.7	1.4 to 5.0	0.004
**Graft function decline**				
Univariate	1.0 (Ref)	1.6	1.0 to 2.6	0.038
Model 1	1.0 (Ref)	1.8	1.1 to 2.8	0.018
Model 2	1.0 (Ref)	1.6	1.0 to 2.6	0.043
Model 3	1.0 (Ref)	1.3	0.8 to 2.1	0.368
Model 4	1.0 (Ref)	1.4	0.8 to 2.4	0.223

Cox proportional hazards regression analysis. Data are presented as hazard ratio and 95% confidence interval (CI) for the two aorto-iliac CaScore groups (low and high). Model 1: adjusted for transplant centre and time between computed tomography and transplantation; model 2: adjusted for model 1 plus donor gender, donor type (living donation, donation after circulatory death (DCD), cold ischemia time, no. of human leukocyte antigen (HLA) mismatches, recipient gender, diabetes mellitus, dialysis vintage, and no. of previous transplantations, statin use; model 3: adjusted for model 2 plus recipient age; model 4: adjusted for model 3 plus donor age.

## Data Availability

The data presented in this study are available on request from the corresponding author. The data are not publicly available due to privacy regulations.

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
