# Peer review of "Aorto-Iliac Artery Calcification and Graft Outcomes in Kidney Transplant Recipients"

_jcm, 2021, doi:10.3390/jcm10020325_

Round 1
Reviewer 1 Report
The paper by Benjamens et al estends their previous work on vacular calcification before kidney transplantation and posttransplant cardiovascular outcomes
Major limitations:
The authors report on average 3 year follow up, but only Focus on 1 year graft function. Please report and analyze 2 and 3 year graft function as well
for their analyis on 1 year graft function 85/547 patients are missing (15,5%!!) why? how complete was follow-up? how many data were missing for the Analysis of graft function decline? how many patients were lost to follow-up?
As informative drop-outs such as graft loss provide important Information regulatory authorities require adequate imputation Methods for GFR Analysis and graft function decline Analysis. The Primary Analysis should always focus on a complete data set with adequate imputation.
the authors arbitrarily devided patients into two Groups (< and > 5600 Ca Score) why? why not median? why not tertiles or quartiles? As continuos variables always provide more information especially after including potential confounders, several models failed to demonstrate a significant impact of calcification on Outcome. As a consequence the random classification results is misleading Graphs e.g. figure 2, figure 4 figure S" and figure S4. While the Graphs suggest a highly significant Impact of calcification, the adjusted models did not Show an Impact. please delete and/or adjust the reporting.
Patients with a high calcification had more deceased donors, especially more donors after cardiac death. Surprisingly, this confounder (donation after cardiac death) was not taken into account. From my understanding only deceased donation vs living donation was included in the models
minor Limitation: instead of using the unusual wording with a negation "non death-censored graft Survival" they should use the Standard wording "Overall graft Survival (incl death)" or " patients alive with a functioning graft"
Reviewer 2 Report
This is by far one of the largest study looking at pre-transplant aorto-iliac calcification and outcomes in kidney transplant recipients. It addresses some of the knowledge gap in this area. I would be grateful if the authors can clarify some of the issues identified in this manuscript.
- Abstract line 35-36, the authors report that pre-transplant aorto-iliac calcification is associated with one-year eGFR in univariate. By the results, it is understood the authors should specify that they are inversely associated
- In the Study design section, please describe if computed tomography scan is a part of the standard workup in kidney transplant recipients or it is done in selected high-risk patients. If so, explain the possible bias in the discussion section
- In line 85, the sentence "death with a functioning graft" is repeated
- In the Short-term Graft Outcomes Results section, Could the authors report the primary nonfunction and vascular complications (Bleeding requiring reoperation, renal artery stenosis, renal artery thrombosis)?
- In line 165, explain why Mean eGFR at one-year after transplantation is measured in 462 patients, when it's described 516 patients at risk one-year after transplantation in Kaplan-Meier survival curve in figure 2
- In line 225, clarify "showed a lower time-to-graft failure" or "showed a lower time-to-death"
- In the Results section, could the authors describe the causes of death?
- In the Table 1, please add "(DS)" in values of eGFR at six-months and one-year
- In the discussion section, please explain the differences found in the results between living donors and deceased donors
Reviewer 3 Report
The authors showed that aorto-iliac artery calcification predicts non-renal outcomes.
I have 1 question.
Q1: Use of statins and warfarin, and blood Mg2+ concentration could affect vascular calcification. Especially, statins increase calcification, but reduce cardiovascular events. How about these variables before transplantation?
Author Response
Rebuttal JCM-1033693
Title: Aorto-Iliac Artery Calcification and Graft Outcomes in Kidney Transplant Recipients
Reviewer #3
The authors showed that aorto-iliac artery calcification predicts non-renal outcomes. I have 1 question: Use of statins and warfarin, and blood Mg2+ concentration could affect vascular calcification. Especially, statins increase calcification, but reduce cardiovascular events. How about these variables before transplantation?
Reply: We thank the reviewer for his/her additional question. We agree that the use of statins, warfarin and blood Mg2+ concentration are interesting variables when addressing vascular calcification. For this specific cohort, warfarin use and blood Mg2+ concentration were not available; however, we were able to include the use of statins. We have made changes in all models accordingly.
Methods: In model 2, we adjusted for model 1 plus donor gender, donor type (living donation, DCD or DBD), cold ischemia time, number of HLA mismatches, recipient gender, diabetes mellitus, smoking, dialysis vintage, number of previous transplantations, statin use pre-transplantation.
Round 2
Reviewer 1 Report
While the manuscript has been improved, major problems remain. The authors did not adequately respond to several important criticisms.
Comment 1: while the authors have no problem with 15% missing data, for renal function analysis they feel it is critical if 25% of data are below 2 years. for 2 and 3 year data a complete case analysis (with all its limitations) would be perfect (and reliable) as 75% and more than 50% (still kore than 270 patients!!) provide data for the 2- and 3-year endpoint. in the transplant literature we have a huge number of manuscripts describing the short-term outcome, but the real problem are long-term outcomes. Therefore I insist on the importance of 2- and 3-year data.
Comment 2 and 3: while the authors argue that complete data analysis is more reliable, they forgot to answer my questions: why patient data were missing? how many were lost to follow-up? how many data were missing for graft function decline? I would strongly recommend a flow chart for the 3 year follow-up with number of patients, who died, were lost to follow-up or had missing values. In addition, for analysis of graft function decline, only 1 GFR value has to be imputed. While imputation is not always the answer, it is mandatory for regulatory purpuses for good reasons. If only patients with full data sets (=patient is alive with a functiong graft and is compliant) are analyzed, a huge bias becomes imminent!! The authors can decide, whether they perform the analysis with imputation as primary analysis or as confirmatory analysis, but imputation is essential for reliable results. To exclude a patient from analysis just because of a missing PTH value, while all other values are present is far from clinical reality, which we want to improve with our reasearch. There a numerous imputation methods, which can provide robust and reliable data. Such an analysis provides some picture of the reality, while a complete case analysis provides another angle of reality and together the picture comes closer to reality.
comment 4: thank you for your explantation, which should be mentioned with some more detail in the text (eg. was derived from the highest tertile ....). As some models provided no significant impact of calcification the dichotomous presentation is misleading as pointed out in my comment. As a consequence the graphs (on GFR decline and death-censored graft survival: figure 2, figure 4, figure S2 and figure S4) should be deleted. The discussion should be amended, as the negative result is important to be clearly discussed. Obviously, calcification only is important for patient survival, which is an important message.
comment 5 and 6: thank you for address my questions adequately
